# The proximity of aluminium atoms influences the reaction pathway of ethanol transformation over zeolite ZSM-5

Kinga Gołąbek [1], Edyta Tabor [2], Veronika Pashkova [2], Jiri Dedecek [2], Karolina Tarach [1] & Kinga Góra-Marek [1✉]

The organization of aluminium atoms in zeolites affects their catalytic properties. Here we demonstrate that the aluminium distribution is a key parameter controlling the reaction pathway of acid catalysed reactions over ZSM-5 zeolites. We study ethanol transformation over two ZSM-5 samples with similar Si/Al ratios of ~15, and with aluminium atoms located mainly at the channel intersections but differently distributed in the framework. One of the samples contains mostly isolated aluminium atoms while the other has a large fraction of two aluminium atoms located in one ring. The FT-IR time-resolved operando study, supported by catalytic results, reveals that the reaction pathway in ethanol transformation over ZSM-5 is controlled by the proximity of aluminium atoms in the framework. ZSM-5 containing mostly isolated Al atoms transforms ethanol in the associative pathway, and conversely ZSM-5 containing a dominating fraction of two aluminium atoms in one ring transforms ethanol in the dissociative pathway.

[1] Faculty of Chemistry, Jagiellonian University in Kraków, Gronostajowa 2, 30-387 Kraków, Poland. [2] J. Heyrovský Institute of Physical Chemistry of the CAS, Dolejškova 2155/3, 182 23 Prague 8, Czech Republic. ✉email: kinga.gora-marek@uj.edu.pl

Zeolites represent the widest and most important group of heterogeneous catalysts applied in oil processing, petrochemistry, synthesis of bulk chemicals and many other fields[1,2]. Moreover, there are highly promising emerging applications of zeolites in biomass transformation, utilization of renewables and the transformation of natural gas into liquids. However, successful application of zeolites in these new fields, as well as the transformation of the current chemical industry or the new raw materials to sustainable ones, requires a new generation of highly active, selective and stable catalysts. Ethanol as a green raw material which can be used in the petrochemical industry as starting material for the production of olefins ($C_3$–$C_4$) and aromatics (benzene, toluene and xylene)[3–5]. Zeolites, because of their acidic properties—originating in the presence of Al atoms—and porous structures, have been widely studied as catalysts for these reactions[3,6,7]. Recently, it was shown that by controlling Al organization in the zeolite framework, it is possible to tune zeolite catalytic properties for the development of new catalysts and processes[6–13].

Zeolites are crystalline, microporous aluminosilicate molecular sieves built from $SiO_4$ and $AlO_4^-$ tetrahedral sharing corners, and forming a regular channel/cavity system with all tetrahedrals located at the channel's surface[11]. The isomorphous Al distribution in the zeolite framework results in its negative charge, which has to be compensated by extra-framework cationic species[6,9,11,14]. Compensation of negative charge by a proton leads to the formation of zeolitic Brønsted acid sites (Si(OH)Al), which become active sites in acid-catalyzed reactions[1,2,10–12]. Thus, Al organization in the zeolite can dramatically affect the properties of zeolite as catalysts. For Si-rich zeolites (Si/Al > 8), the presence of isolated Al atoms (one Al atom in one ring) and Al pairs (Al–O–(Si–O)$_2$–Al sequences in one ring) has been reported (Fig. 1)[6,11].

The distribution of Al atoms in Si-rich zeolites is neither random nor controlled by simple rules, but depends on the synthesis conditions and can be modified to a wide extent. This opens the possibility to tune the properties of zeolite catalysts for individual reactions[6,11]. The control of Al distribution in zeolites by the synthesis method was managed recently; thus, there is a strictly limited number of papers dealing with the influence of Al distribution in zeolites on the acid-catalyzed reactions. Moreover, except Al distribution, the location of Al atoms in the zeolite channels is a parameter that could affect catalytic performance of zeolite catalysts. Thus, up to now, the effect of Al distribution in zeolites on acid-catalyzed reactions was studied in the transformation of methanol to olefins (MTO) over SSZ-13 zeolite[7,15,16] and propene oligomerization over ZSM-5[12,13]. The study of the influence of Al distribution in SSZ-13 on catalytic performance in MTO reaction was enabled because SSZ-13 zeolite has only one framework Al T site. However, in the case of ZSM-5 structure with 24 T sites, the progress in the $^{27}$Al MQ

MAS NMR (multiple-quantum magic-angle spinning nuclear magnetic resonance) analysis allows for characterizing Al siting in ZSM-5 zeolites, and thus the influence of Al location in ZSM-5 on acid-catalyzed reactions could be determined[12,13]. The $^{27}$Al MQ MAS NMR results showed that in ZSM-5 synthesized using tetrapropylammonium cation as a structure-directing agent, Al atoms are predominantly located at the channel intersections independently of the concentration of single Al atoms and Al pairs[17].

This study analyzes the effect of the distribution of Al atoms in the framework of H-ZSM-5, and thus of defined proximity of charge-balancing protons on the process of ethanol transformation to hydrocarbons, including adsorption and desorption steps. For the purpose of this study, the H-ZSM-5 catalysts with similar Si/Al molar ratios containing predominantly (i) single Al atoms (83% of Al atoms observed as single Al atoms)—H-ZSM-5$_{(s)}$ (Si/Al = 15) and (ii) Al pair (70% of Al atoms observed as Al pairs)—H-ZSM-5$_{(p)}$ (Si/Al = 14.5) are hydrothermally synthesized. Complete analysis of the state and location of framework Al atoms regarding channel intersections and their distribution between Al pairs and single Al atoms is performed using $^{27}$Al (3Q) and $^{29}$Si MAS NMR spectroscopy, Co(II) ion-exchange capacity, and visible (Vis) spectroscopy of bare Co(II) ions in dehydrated Co zeolites[12,13]. The novelty of this work lies with the direct analysis of the effect of aluminium organization in the zeolites on the reaction pathways in acid-catalyzed reactions.

## Results and discussion

**Mechanism of ethanol dehydration over ZSM-5.** According to the research reports, the dehydration of ethanol to diethyl ether is represented by two different pathways, termed the associative pathway and the dissociative pathway[7,18,19]. In the associative pathway, an ethanol molecule is adsorbed at a proton site to form a hydrogen-bonded ethanol monomer (Fig. 2A); then by the adsorption of another alcohol molecule in close vicinity, the dimeric species are formed (Fig. 2B). In the dimeric intermediate, the proton is held between the alcohol oxygen atoms. Subsequently, protonated dimers rearrange and decompose into water and an adsorbed diethyl ether (Fig. 2C), which desorbs to regenerate the proton site. The presence of protonated ethanol dimers during ethanol dehydration on zeolitic catalysts was confirmed previously[20,21]. Similarly, the dissociative pathway involves the formation of a hydrogen-bonded ethanol monomeric intermediate (Fig. 2A). By water molecule elimination, this intermediate is transformed to an ethoxy group[22] replacing the proton in the Si(O$^-$)Al site (Fig. 2D). Further transformation of the ethoxy group can either provide ethylene while a Brønsted acid site is regenerated (Fig. 2E), or ethanol–ethoxy pair is formed by adsorption of second ethanol on an adjacent-framework oxygen (Fig. 2F). The ethanol–ethoxy pair forms diethyl ether (Fig. 2G), which desorbs to regenerate the Brønsted acid site.

**Synthesis protocol and physicochemical characterization.** The distribution of Al atoms in ZSM-5 structure is neither random nor controlled by statistical rules, but it can be tuned by the synthesis method[6,11,23]. Thus, H-ZSM-5$_{(s)}$ and H-ZSM-5$_{(p)}$ with close concentrations of framework Al atoms (Si/Al ~ 15, Table 1), but substantially differing in the distribution between Al pairs (AlSiSiAl sequences in 6MRs) and far distant single Al atoms (AlSi$_{n>2}$Al in different rings), were hydrothermally synthesized (see refs. [24,25]). Both H-ZSM-5$_{(s)}$ and H-ZSM-5$_{(p)}$ were prepared using silica sol as a source of silica, and synthesis was carried out in the presence of TPAOH (tetrapropylammonium hydroxide) as a structure-directing agent to obtain Al atoms located mostly at the channel intersections. The synthesis protocol differed in the

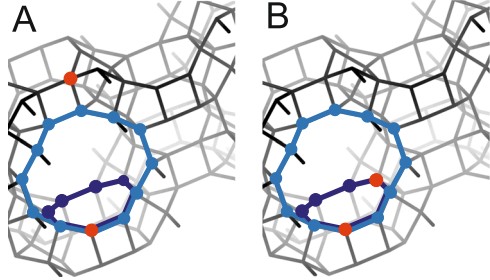

**Fig. 1 Representation of Al organisation in zeolite framework model.**
Isolated aluminium atoms (**a**) and pairs of aluminium atoms (**b**); aluminium atoms marked in red.

**Fig. 2 Possible pathways of ethanol dehydration over zeolites.** The particular steps in dissociative and associative reaction pathways in ethanol dehydration over ZSM-5. Transition species and the final products formed over ZSM-5 after interaction with ethanol are depicted in this figure: hydrogen-bonded ethanol monomer (A), dimeric ethanol species (B), diethyl ether and water (C), ethoxy group (D), desorbed ethylene (E), ethanol–ethoxy pair (F) and diethyl ether (G).

**Table 1 Physicochemical characterization of H-ZSM-5$_{(s)}$ and H-ZSM-5$_{(p)}$.**

| Material | Si/Al[a] | Al$_{ICP}$[a] | S$_{micro}$[b] | S$_{BET}$[b] | V$_{micro}$[b] | V$_{total}$[b] | Co$_{max}$/Al | Single Al | Al pairs | α[c] | β[c] | Al (H$^+$) at intersections |
|---|---|---|---|---|---|---|---|---|---|---|---|---|
| | | (µmol g$^{-1}$) | (m$^2$ g$^{-1}$) | | (cm$^3$ g$^{-1}$) | | molar | % | | | | |
| ZSM-5$_{(s)}$ | 15.0 | 900 | 425 | 440 | 0.18 | 0.21 | 0.12 | 83 | 17 | ND | ND | >95 |
| ZSM-5$_{(p)}$ | 14.5 | 995 | 404 | 419 | 0.17 | 0.20 | 0.35 | 30 | 70 | 37 | 54 | ~65 |

[a]Calculated from chemical analysis.
[b]Low-temperature N$_2$ adsorption/desorption measurements.
[c]Portion of Al(H$^+$) at channel intersections as determined for H-ZSM-5$_{(s)}$ from $^{27}$Al 3Q MAS NMR spectra, and for H-ZSM-5$_{(p)}$ from Al concentrations in α- and β-rings from Co Vis spectra (Fig. 8).

Al source: to prepare H-ZSM-5$_{(p)}$ and H-ZSM-5$_{(s)}$, aluminium butoxide or polyaluminium chloride was used, respectively.

Distribution of Al atoms in zeolites was estimated using the well-proven methodology of the titration of Al pairs with [Co(H$_2$O)$_6$]$^{2+}$ and ultraviolet–Vis (UV–Vis) characterization[6,7,10,12,13]. The maximum exchange level of [Co(H$_2$O)$_6$]$^{2+}$ ions (Co$_{max}$) without the occurrence of any hydrolytic Co(II) complexes (see ref. [26]) reflects the concentration of Al pairs. Siting of Al atoms with the respect to the channel crossing and location in the zeolite channels was analyzed in the case of the sample with single Al atoms using $^{27}$Al 3Q MAS NMR, for details see ref. [17]. In the case of ZSM-5 structure, the exact occupation of the framework T sites by Al atoms cannot be analyzed using this method due to the high number of T sites. Thus, $^{27}$Al 3Q MAS NMR spectroscopy was employed to evidence or exclude the location of Al atoms in the T sites not related to the channel crossing (note that <17% of T sites is not located at the ZSM-5 intersections). It was shown that in H-ZSM-5$_{(s)}$ exclusively, T sites located at the intersections are occupied by Al atoms[17]. Moreover, $^{27}$Al 3Q MAS NMR cannot be employed for the analysis of Al siting in samples containing a significant fraction of Al pairs. The vicinity of other Al atoms dramatically influences the geometry and NMR parameters[27]. Thus, to determine the concentration of Al pairs in ZSM-5, Co(II) ions were used as a probe[6,12,13,23]. UV–Vis spectroscopy results confirmed that in the dehydrated Co-ZSM-5, bare Co(II) ions balance two close Al atoms in different rings, which form different cationic sites denoted in ZSM-5 as α, β and γ sites. Al atoms of the α and β sites are located at the channel intersection, and their concentration represents minimum concentration of Al atoms at the channel intersection in the sample (Table 1). Siting of Al atoms

in the γ site is below 10%, and siting of single Al atoms cannot be estimated in H-ZSM-5$_{(p)}$.

UV–Vis and $^{27}$Al 3Q MAS NMR spectroscopic results confirmed that both prepared ZSM-5 samples containing the majority of Al atoms at channel intersection but they differed only in Al organization in the framework. While ZSM-5$_{(s)}$ contains single Al atoms mainly (83% of Al as single Al atoms), the zeolite ZSM-5$_{(p)}$ possesses Al in pairs (70% Al in pairs)[12,17] (Fig. 1 and Table 1). The studied samples exhibited highly crystalline structures, as confirmed by powder X-ray diffraction[6,24] (Fig. 3a). The isotherms of ZSM-5$_{(s)}$ and ZSM-5$_{(p)}$ were representative for purely microporous structures (Table 1)[28]. The value of the specific area (419–440 m$^2$ g$^{-1}$) and micropore volume (0.17–0.18 cm$^3$g$^{-1}$) were also typical of 10-ring zeolites[12,26] (Table 1). Both prepared ZSM-5 of mono-modal but different Al distributions were subjected to operando Fourier-transform infrared (FT-IR) spectroscopic evaluation coupled with gas chromatography–mass spectrometry product analysis in terms of the nature, stability and efficiency in the ethylene and higher hydrocarbons yield. The location of Al atoms in tetrahedral positions guarantees the formation of the Si(OH)Al groups as the only species (Fig. 3b), while a negligible amount of Lewis acid sites was detected in $d_3$-acetonitrile adsorption studies (Table 2, experiment details are described in Methods section). $d_3$-Acetonitrile is highly suitable for quantification of acid sites in the constrained zeolite channel system. Nevertheless, its applicability for the analysis of the acid strength of the active sites in the zeolites is limited. Thus, pyridine (Py) sorption analysis for both zeolites ZSM-5$_{(p)}$ and ZSM-5$_{(s)}$ (Table 2 and Fig. 3c) was employed to determine the strength of acid sites. The results of

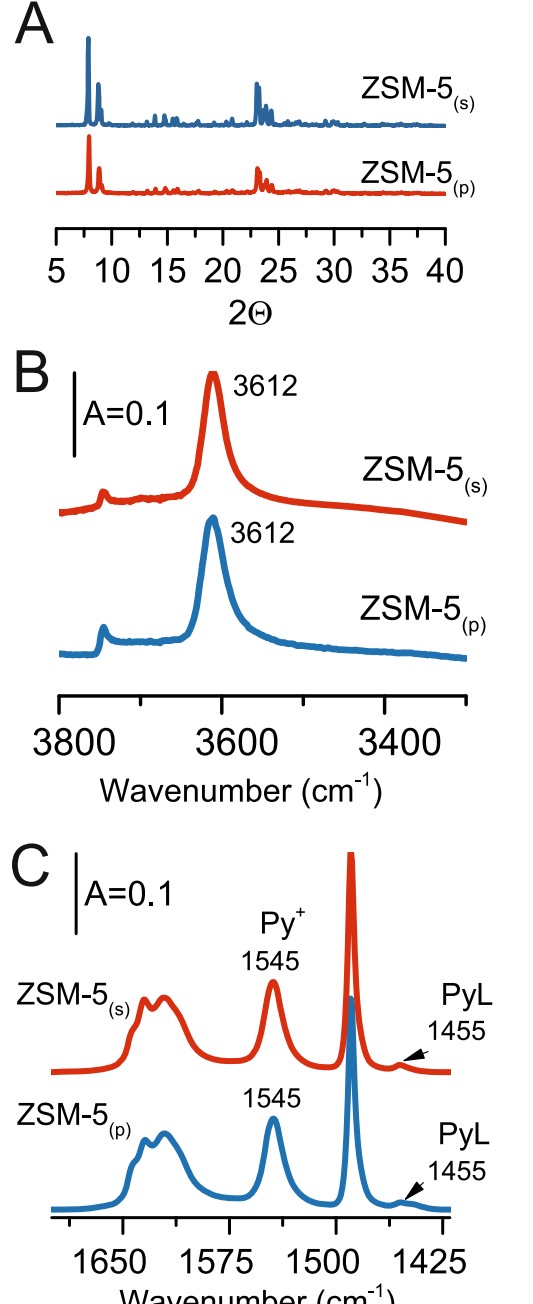

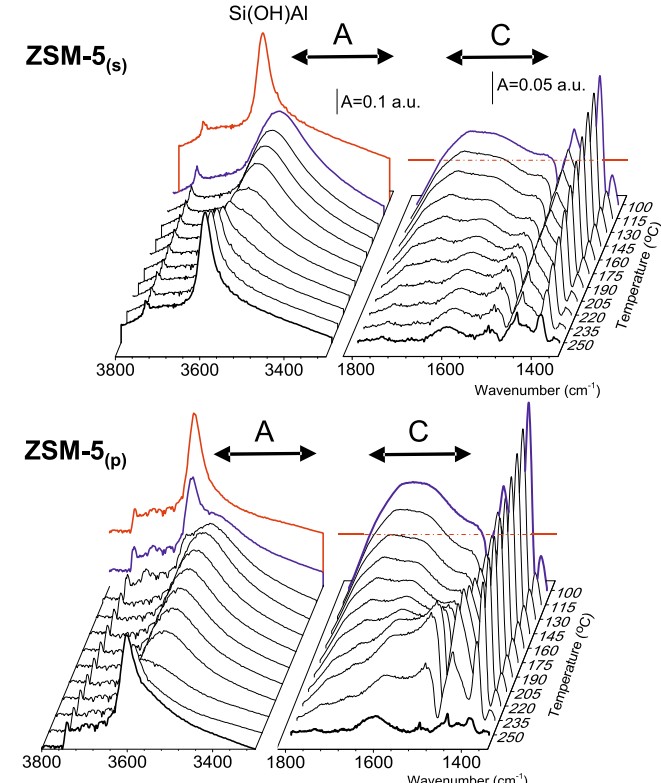

**Fig. 4 FT-IR spectra of zeolites interacting with $C_2H_5OH$ at various temperatures.** FT-IR spectra of ZSM-5 after 60 min at 550 °C followed by 10 min at 100 °C in $N_2$ (red line), 20s interaction with ethanol at 100 °C (blue line) and upon the contact with ethanol at temperature from 100 to 250 °C within 4 min (black lines). Component C is presented as the difference between the spectrum collected in the sequence described above and the spectrum of activated zeolite.

**Fig. 3 Characterization.** XRD (**a**) and FT-IR characterization: Si(OH)Al groups (**b**) and Py adsorbed at 170 °C (**c**).

**Table 2 Concentration and strength of Brønsted (B) and Lewis (L) acid sites.**

| Material | Si/Al | $B^a$ ($\mu$mol g$^{-1}$) | $L^a$ | $B^b$ | $L^b$ | Acid strength | |
|---|---|---|---|---|---|---|---|
| | | | | | | $B^c$ | $L^c$ |
| ZSM-5$_{(s)}$ | 15.0 | 886 | 12 | 899 | 22 | 0.95 | 0.62 |
| ZSM-5$_{(p)}$ | 14.5 | 894 | 25 | 926 | 36 | 0.97 | 0.76 |

$^a$Calculated from acetonitrile adsorption.
$^b$Calculated from pyridine adsorption.
$^c$The strength of B and L sites (expressed by Py$_{330}$/Py$_{170}$) calculated from quantitative IR studies of Py sorption.

Py sorption are in line with the results from $d_3$-acetonitrile adsorption, they confirmed low concentration of Al atoms with Lewis character (Table 2). Moreover, it was shown that both zeolites provide protonic sites with similar acid strength (Table 2), pointing to a minor influence of neighbour Al atom in Al pairs on the Brønsted acid sites strength[14]. It should be however underlined that for small molecules conversions (e.g. ethanol, etylene, ect.) the stabilization of the adsorption state by the intimate confinement forced by the paired proton (the proximity of Al atoms) cannot be neglected.[6,7,12,13]. The importance of the confinement effects arising from neighbouring protonic sites was herein discussed for ethanol dehydration on ZSM-5 with on-purpose tuned Al distribution within the framework.

**Rapid scan FT-IR studies**. The IR spectrum of both ZSM-5$_{(s)}$ and ZSM-5$_{(p)}$ recorded at 100 °C after treatment in nitrogen at 550 °C exhibited the 3612 cm$^{-1}$ band of the Brønsted acid sites (Si (OH)Al) (Fig. 4, red line). Upon contact with ethanol at 100 °C, in an amount corresponding to one alcohol molecule per one (Si (OH)Al) (blue line), the Si(OH)Al sites of ZSM-5$_{(s)}$ are erased. At 175 °C, the Si(OH)Al band of ZSM-5$_{(s)}$ starts to recover and finally restores its initial intensity at 220 °C (Fig. 4). In contrast, the vast majority of the Brønsted acid sites of ZSM-5$_{(p)}$ is not involved in the interaction with ethanol at 100 °C; still, ca. 80% of protonic sites remain unperturbed (blue line). The intensity of the Si(OH)Al band of ZSM-5$_{(p)}$ gradually decreases up to total consumption of Brønsted acid sites at 160 °C (Fig. 4). The changes in the intensity of the Brønsted acid sites band of both ZSM-5$_{(s)}$ and ZSM-5$_{(p)}$ depending on the temperature interaction with ethanol

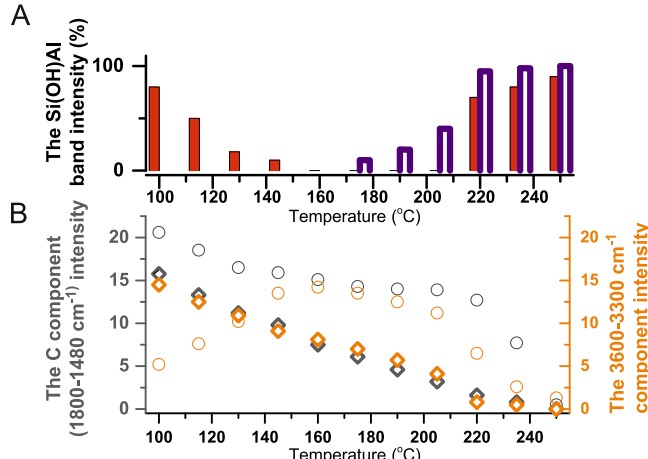

**Fig. 5 Intensity of FT-IR band evolution in 100–260 °C temperature range.** Evolution of the band intensity of the Si(OH)Al groups (**a**) and H-bond components at the regions: 1800–1480 and 3600–3300 cm$^{-1}$ with temperature over ZSM-5$_{(s)}$ (empty symbols) and ZSM-5$_{(p)}$ (full symbols) (**b**).

are presented in the upper part of Fig. 5. In ZSM-5$_{(p)}$, the Si(OH)Al is recovered at 220 °C, that is, at significantly higher temperature than in ZSM-5$_{(s)}$. Finally the Si(OH)Al band exhibits its initial intensity at 250 °C (Fig. 5, upper part). This difference in the reactivity towards ethanol between two ZSM-5 studied provides an argument for constrained accessibility of protonic sites for adsorbates in ZSM-5$_{(p)}$. Once the Brønsted groups were consumed, no depletion of the weaker-acidity silanols was observed, which confirmed that ethanol interacts with Si(OH)Al sites exclusively. The decreasing intensity of the Si(OH)Al band (3612 cm$^{-1}$) in ZSM-5$_{(s)}$ and ZSM-5$_{(p)}$, due to the interaction with ethanol, is accompanied by the appearance of (i) a new down-shifted structure of hydrogen-bonded ethanol at 3600–3300 cm$^{-1}$ (the A component) and 1800–1480 cm$^{-1}$ (the C component), and (ii) the CH$_3$-group vibration band (CH$_3$ degenerate bending mode at 1446 cm$^{-1}$ and CH$_3$ symmetric bending at 1395 cm$^{-1}$) (Fig. 4)[7,19]. The CH$_2$ bending modes are too weak to be detected. Considering dependence of the A and C components intensities on the temperature of ethanol interaction with both zeolites (Fig. 5, lower part) it can be concluded that in zeolite ZSM-5$_{(s)}$ the A and C components exhibit the same intensity, whereas a significantly higher intensity of the C component in ZSM-5$_{(p)}$ is observed (Fig. 4). This spectral feature shows that the hydrogen bond of the medium–strong strength is formed in the zeolite with a majority of single Al atoms (ZSM-5$_{(s)}$): the proton is located in the middle of two oxygen atoms originating from Si(O)Al and ethanol molecule. In contrast, in the sample with a large fraction of Al pairs (ZSM-5$_{(p)}$), very strong H–bonds are observed. Both the presence of the Al–O–(Si–O)$_2$–Al units in ZSM-5$_{(p)}$ and enhanced basicity of neighbouring-framework oxygens result in strong stabilization of ethanol molecules, the most probably in the form of the protonated species Si(O–)Al···($^+$HOH)C$_2$H$_5$[18,29–31]. When temperature increases, the linear decrease in the intensity of the broad bands describing H–bonding is observed (see Fig. 5 upper part). Conversely, for ZSM-5$_{(p)}$ when the temperature rises to 160 °C, the intensity of the H-bonded ethanol components A and C upsurges, in line with the erosion of the Si(OH)Al species (Figs. 4 and 5).

Further, in spite of the same intensity of the C component (Figs. 4 and 5) in ZSM-5$_{(p)}$, the rearrangement in strongly hydrogen-bonded moieties is detectable in the temperature range

of 160–205 °C. From this point (205 °C), a linear decrease in the intensity of the C component is observed up to the initial intensity of Brønsted acid sites band restoration. It is clearly shown that the presence of single and paired protons influences the formation of H-bonded species representative for the active sites in dehydration of ethanol, therefore strongly pointing to the impact of Al distribution in the zeolite matrix on the reaction mechanism. The pore confinement effects largely contribute to transition-state stabilization determining the reaction mechanism; thus both associative and dissociative pathways are active in the whole temperature range. Recently, DFT studies[32] confirmed the correlation between the reaction temperature of transition from associative to the dissociative pathway in ethanol dehydration over the CHA structure. Herein, we discuss the influence of Al siting on ethanol dehydration over ZSM-5$_{(s)}$ and ZSM-5$_{(p)}$ followed at 250 °C in rapid scan IR experiments (Fig. 6). Both catalysts interacted with ethanol at 250 °C; however the H-bonded monomeric ethanol species that existed in both ZSM-5$_{(s)}$ and ZSM-5$_{(p)}$ started to decompose at different rates (Fig. 5, lower part). The surface of ZSM-5$_{(s)}$ is free from ethanol-originated species at 250 °C, showing that all these species are easily released from single Al atoms (Fig. 6, red line). It suggests that dehydration of ethanol over ZSM-5$_{(s)}$ occurs mainly on the associative pathway. In contrast, at 250 °C the spectrum of ZSM-5$_{(p)}$ is free of Fermi resonance spectral features (the hydrogen-bonded species are absent); however still, the ethanol-originated band at 1395 cm$^{-1}$ is easily detected (Fig. 6, red line). Under these conditions, the presence of the 1395 cm$^{-1}$ band is indicative for the ethoxy species –OC$_2$H$_5$ formed due to strong stabilization of one ethanol molecule in the form of the protonated species Si(O$^-$)Al···($^+$HOH)C$_2$H$_5$ over ZSM-5$_{(p)}$ and their subsequent dehydration (Fig. 6). Over the paired protons, the co-presence of both H-bonded ethanol species and ethoxy groups is therefore postulated. The co-operative behaviour of protonic sites is evidenced in detail by the 2D COS (two-dimensional correlation spectroscopy) maps obtained from the correlations of two different spectral regions: the 3100–2500 cm$^{-1}$ and 1650–1350 cm$^{-1}$ of C–H stretching and bending vibrations, respectively (Fig. 6). On 2D maps, the overlapping peaks are well resolved, which allows their detecting and characterizing the specific sequential order of spectral intensity changes. The 2D synchronous map presenting correlations between $v_1$ and $v_2$ wavenumbers consists of the positive autopeaks and the positive or negative crosspeaks. The positive crosspeaks represent the simultaneous changes in the intensities of the $v_1$ and $v_2$ bands, that is, their simultaneous increase or decrease. The negative crosspeaks represent the band intensity changes at the expense of one to the other. The negative correlations between the Si(OH)Al band (3612 cm$^{-1}$) and hydrogen-bonded species (the C component at ca. 1520 cm$^{-1}$) and CH$_3$ vibration bands (1445 and 1395 cm$^{-1}$) point to the involvement of only those species in ethanol conversion over ZSM-5$_{(s)}$. No correlation peaks are found for other surface species or products providing an argument for their coincidental formation, if any. Besides, the most vital features found in 2D COS maps for zeolite ZSM-5$_{(p)}$ are the correlations between the ethoxy groups (1395 cm$^{-1}$) and hydrogen-bonded but non-protonated ethanol molecules (the 3530 cm$^{-1}$ band)[33]. The simultaneous releasing both an ethoxy groups from the Si(OH)Al sites and non-protonated hydrogen-bonded ethanol molecules is documented by the positive sign of the (3350 cm$^{-1}$ × 1395 cm$^{-1}$) correlation. The reaction between ethoxy intermediates and a second ethanol molecule adsorbed at adjacent-framework oxygen, is indicative for a surface ethanol–ethoxy pair-mediated routes in ZSM-5$_{(p)}$[34].

On the other hand, the associative route of the ethanol dehydration is facilitated on Brønsted sites provided by isolated Si

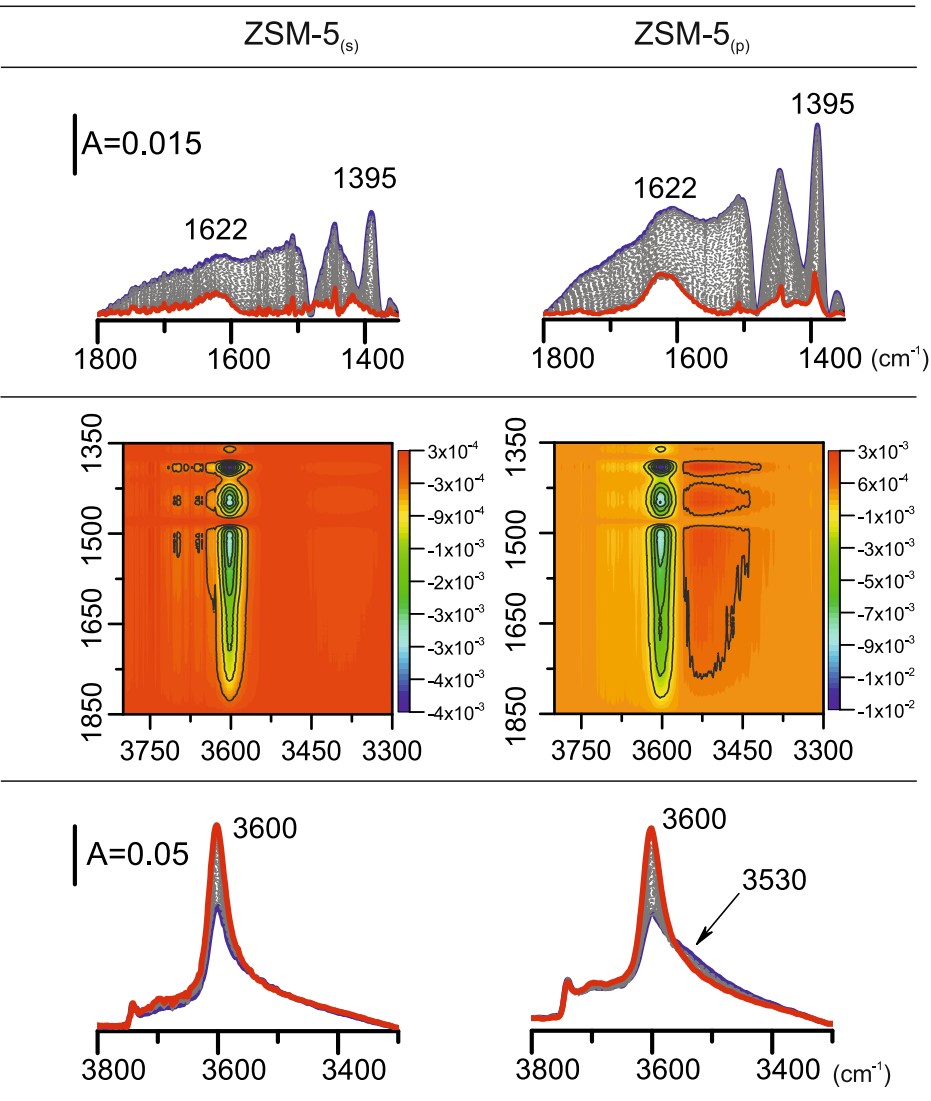

**Fig. 6 FT-IR spectra and 2D maps for C₂H₅OH transformation at 250 °C.** The 2D correlation maps for ethanol transformation at 250 °C on the zeolites studied derived from spectra collected in rapid scan mode (one spectrum collected in 1 s). Blue line—spectrum collected after 2 s of interaction with ethanol; red line—spectrum after 4 min of interaction with ethanol at 250 °C.

(OH)Al hydroxyls because solely hydrogen-bonded species are involved in the ethanol transformation over ZSM-5(s). The strong impact of Al siting on the dissociative ethanol dehydration pathway is therefore clearly proved. Rapid scan FT-IR studies demonstrating that the adsorption of ethanol on the protonic site proceeds faster on isolated protonic sites than on Al pairs are supported by operando study results. Independently from aluminium atoms distribution, at 250 °C both catalysts transform ethanol molecules to ethylene exclusively; however, the inhibited conversion of ethanol is observed in the presence of Al pairs (Fig. 7).

**Operando GC and FT-IR study**. Operando study results (GC and FT-IR) revealed that at temperature of 250 °C, higher ethanol conversion on ZSM-5(s) than on ZSM-5(p) (89% and 65% for ZSM-5(s) and ZSM-5(p), respectively) occurred, see Fig. 7. At 250 °C, on both studied ZSM-5 samples, ethanol is transformed to ethylene (dominant fraction) and C₂–C₆ and traces of aromatics are detected. In zeolite ZSM-5(p), ethylene formation becomes significantly accompanied by C₄ fraction (the selectivity is

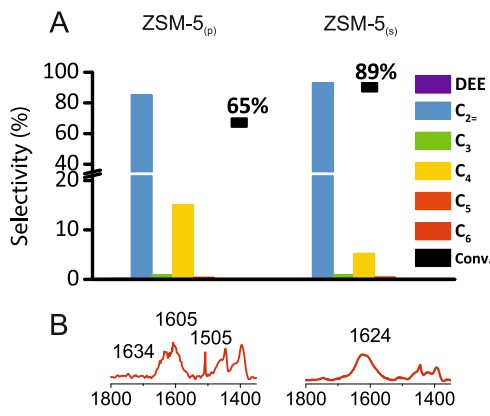

**Fig. 7 Operando FT-IR and GC results.** Catalytic characteristics (**a**) and FT-IR spectra in the deformation C–H vibration region (**b**) collected after 4 min ethanol transformation at 250 °C.

15%)[35]. The promoted dimerization is also manifested by the presence of the $1624 \, cm^{-1}$ complex band of conjugated C=C systems (Fig. 7, red line)[35]. It clearly shows that skeletal ethoxyls formed in ZSM-5$_{(p)}$ readily participate in the formation of C–C bonds. Besides, the rapid hydrogenation of C$_{4=}$ to C$_4$ observed in substantial extent in the presence of paired protons on ZSM-5$_{(p)}$ (C$_{4=}$/C$_4$ 0.33 and 0.06 for ZSM-5$_{(s)}$ and ZSM-5$_{(p)}$, respectively) and the formation of aromatic intermediate species (*m*-xylene: the $1605 \, cm^{-1}$ band and 1,2,4-trimethylbenzene: the $1505 \, cm^{-1}$ band)[36,37] (Fig. 7) points to exceptional activity for hydrogen-transfer processes[7,18]. Enhanced oligomerization was also reported in 1-butene[18] and propene[4,7] reactions on ZSM-5 with Al pairs. Bleken et al.[38] have tested 10-ring topologies (IM-5, TNU-9, ZSM-11 and ZSM-5) as catalysts for the methanol to hydrocarbon reaction (MTH), and concluded that the cavity size of 3D 10-ring zeolites is crucial for their stability as MTH catalysts. In this work, we demonstrated that of primary importance is the rate of decomposition of the adsorbed ethanol species, which is significantly inhibited by the strong hydrogen bonding provoked by the presence of Al–O–(Si–O)$_2$–Al sequences. The results of ethanol dehydration reactions studied in this research imply that the design and selection of microporous catalysts for performing shape-selective reactions require us to consider the stability of the surface intermediates as well as the location of Brønsted acid sites.

In this work, by using FT-IR time-resolved operando study, we demonstrated that the Al atom distribution represents a key parameter controlling the reaction pathway of ethanol transformation over ZSM-5. The time-resolved operando studies of ethanol transformation on zeolites showed that the Al pairs of Al–O–(Si–O)$_2$–Al sequences provide strong stabilization for the ethanol molecule by an additional hydrogen bond. The ethanol–ethoxy-mediated dissociative route of ethanol dehydration is therefore significantly facilitated. The Brønsted sites closely located in Al–O–(Si–O)–Al units also promote dimerization and hydrogen-transfer reactions. At higher temperatures, the regeneration of the Si(OH)Al occurs by the release of ethylene. Therefore, the creation of the most favourable ethoxy intermediate transformed directly to ethylene can still be followed by ethanol–ethoxy-mediated dissociative path. The on-purpose tuned distribution of Al atoms, therefore, opens the door for the creation of highly selective tailor-made zeolitic catalysts.

## Methods

**Sample preparation and treatment.** The synthesis procedure was based on preliminary preparation of aluminosilicate precursors[12]. Crystallization of zeolites was driven by ZSM-5 crystals seeding, supported by ethanol and ammonia. For the preparation of aluminosilicate precursor for ZSM-5$_{(p)}$ (Si/Al ≈ 15) synthesis, aluminium butoxide (192.445 g) was mixed with 750 ml of distilled water; to this mixture, a 25% NH$_3$ solution was added while stirring to reach pH = 9. Then, the silica sol (30% of SiO$_2$, 2503.563 g) was added, and the mixture was stirred for 15 min, before mixing with a blender. Finally, the whole slurry was spray-dried giving dry amorphous aluminosilicate precursor. For the hydrothermal synthesis of zeolite, 900 g of the precursor was used with the addition of 1 wt% of seeding ZSM-5 crystals, 1847.2 g of distilled water, 58.0 g of NaOH, 303.4 g of ethanol and 223.9 g of 25% NH$_3$. The whole mixture was transferred into a stirred autoclave, and the crystallization ran at 130 °C for 2 days.

An aluminosilicate precursor for the preparation of ZSM-5$_{(s)}$ (Si/Al ≈ 15) sample was precipitated in a vigorously stirred reactor using the silica sol (3600 g, 30% of SiO$_2$, NH$_3$ stabilized) and polyaluminium chloride PAX-18 (300 g of a solution containing 8.65% Al). From two pumps, the two solutions were continuously dropping into the reactor with 1 L of distilled water, while stirring. The mixture showed pH 3.5, and it was neutralized by the slow dropping of 5% NH$_3$ solution to achieve pH = 9.0. Then, the slurry was stirred for 15 min, filtered without washing and let on a filter overnight. In a second step, sodium hydroxide (101.8 g) was dissolved in 400 g of distilled water, and the wet precursor (2200 g) was added and stirred for 5 min. Thereafter, 225.5 g of ethanol and 166.4 g of 25% NH$_3$ solution were added. The whole slurry was then stirred for 30 min, transferred into the autoclave and finally 1 wt% of ZSM-5 seeding crystals were added. The crystallization was performed at 130 °C for 3 days. After drying and calcination in

air, the sample was transferred to the Na$^+$ form, which was next either maximum Co(II) exchanged for analysis of Al distribution or exchanged with ammonia followed by calcination to obtain the protonic form.

**X-ray diffraction.** Wide-angle X-ray diffraction (XRD) patterns were taken with a Rigaku Multiflex diffractometer equipped with CuKα radiation (40 kV, 40 mA). The XRD patterns evidenced well-developed crystalline material (Fig. 2a).

**Low-temperature nitrogen sorption.** The nitrogen sorption measurements were performed on a Quantachrome Autosorb-1-MP gas sorption system at −196 °C.

**$^{27}$Al (3Q) MAS NMR spectroscopies.** Complex analysis of the state and location of framework Al atoms regarding channels and their distribution between Al pairs and single Al atoms were obtained by employing $^{27}$Al (3Q) and Co(II) ion exchange to determine Al pairs. Information on Al siting of the Al$_{1Al}$ atoms in ZSM-5$_{(s)}$ was obtained from the 2D plot analysis of the $^{27}$Al 3Q MAS NMR spectra of the hydrated Na samples combined with quantitative analysis with the simulation of the $^{27}$Al single-pulse experiment. The resonances with an isotropic chemical shift of 56.7 and 56.2 ppm predominate in ZSM-5$_{(s)}$ (95% of Al). According to our previous study, these values can be attributed to the calculated $^{27}$Al chemical shifts of 56.7, 57.0, 57.1, 57.1 and 57.2 ppm, which were assigned to Al atoms substituted in the T11, T21, T2, T16 and T23 sites, respectively, of monoclinic ZSM-5[6,12]. Only the T16 site of those assignable to the $^{27}$Al NMR resonance at 56.4 ppm is not located at the channel intersection. This together with the fact that both ZSM-5 samples were prepared using TPA$^+$ ions, located at the channel intersections, allows us to suggest that the Al atoms of the $^{27}$Al resonance at 56.4 and 56.0 ppm are most likely present at channel intersections. Thus, the framework Al atom in the ZSM-5$_{(s)}$ sample is mostly (85%) located at the channel intersections.

The $^{27}$Al (3Q) NMR spectra analysis of Al location cannot be employed with ZSM-5$_{(p)}$ containing predominantly Al pairs as close Al atoms significantly affect $^{27}$Al resonances[7,12,27,39]. On the other hand, the zeolite containing prevailingly Al pairs provides advantageously quantitative exchange of Co(II) ions (Co$_{max}$) at the individual Al pair sites, which, in dehydrated Co-zeolites, can be used for the determination of positions of these Co(II) ions in the individual rings and channels. Quantitative analysis of d–d transitions of individual bare Co(II) ions (as probes of charge-balancing Al pairs), obtained from Vis spectra of dehydrated Co(II)-ZSM-5, provided infomation on the location of Al pairs in the framework rings and channel system[26,27,39]. UV–Vis diffuse-reflectance spectra of fully exchanged Co(II)-ZSM-5 samples dehydrated at 480 °C were recorded on a Perkin-Elmer Lambda 950 UV–Vis–NIR spectrometer equipped with integrating sphere for diffuse-reflectance measurements covered by Spectralon®, which also served as a reference. The reflectance was recalculated using the Schuster–Kubelka–Munk function $F(R_\infty) = (1 − R_\infty)^2 / 2R_\infty$, where $R_\infty$ is the diffuse reflectance from a semi-infinite layer and $F(R_\infty)$ is proportional to the absorption coefficient. Refer to details of the d–d vibrations of the individual Co(II) ions and distribution of Al in the MFI framework.

The concentration of Al atoms in Al pairs in individual (α, β, γ) 6MRs was calculated using Eq. (1). Co-oxo species were not detected in Co-ZSM-5$_{(p)}$ and Co-ZSM-5$_{(s)}$[3].

$$Al_{2Al} = 2\,Co_{max} = 2[Co(II)\text{-}\alpha + Co(II)\text{-}\beta + Co(II)\text{-}\gamma]. \quad (1)$$

The concentrations of the individual bare Co(II)-α, Co(II)-β and Co(II)-γ ions were determined from the quantitative analysis of the characteristic d–d transitions of the Vis spectra of dehydrated Co-ZSM-5 zeolites. Deconvolution of the UV–Vis spectrum Co-ZSM-5$_{(p)}$ is shown in Fig. 8 (upper panel).

The concentration of single Al$_{1Al}$ atoms was determined by Eq. (2)

$$Al_{1Al} = Al_{total} − 2\,Co_{max}, \quad (2)$$

where Al$_{total}$ was derived from chemical analysis.

**IR spectroscopy.** The acidic function of the studied zeolites was characterized by FT-IR studies of Py (Pyridine, Sigma-Aldrich, purity ≥ 99.8%) adsorption according to the procedure described in ref. [40]. The concentration of acid sites was attained from the intensities of pyridinium ion band at $1545 \, cm^{-1}$ and Py bonded to Lewis sites at $1455–1445 \, cm^{-1}$ recorded upon the desorption at 170 °C (Fig. 2). The acid strength of protonic sites was measured in quantitative Py-sorption IR experiments. The Py$_{330}$/Py$_{170}$ ratio (where Py$_{330}$ and Py$_{170}$ are the values of the Py IR band intensities upon adsorption at the temperature of 330 °C and 170 °C, respectively) expresses the number of Py molecules neutralizing protonic sites that survived desorption at 330 °C (Table 2). Thus, the acid strength presented in Table 2 is as a dimensionless value. FT-IR spectra of adsorbed acetonitrile on dehydrated ZSM-5 samples were measured after adsorption of 1.3 KPa of d$_3$-acetonitrile) at room temperature (RT) for 30 min, followed by 30 min of evacuation at RT. The bands at 2297 and $2325 \, cm^{-1}$ characterized C≡N interacting with Brønsted and Lewis sites, respectively. The values of absorption coefficients $\varepsilon_B = 2.05 \, cm \, \mu mol^{-1}$ and $\varepsilon_L = 3.62 \, cm \, \mu mol^{-1}$ were taken from ref. [41].

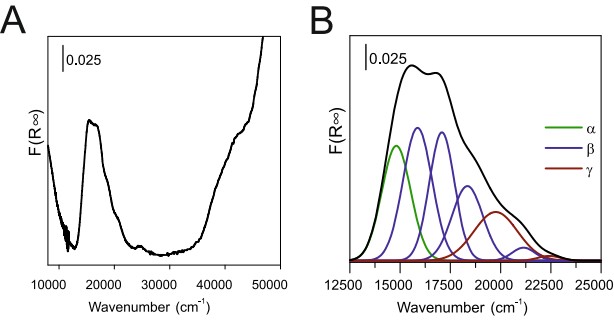

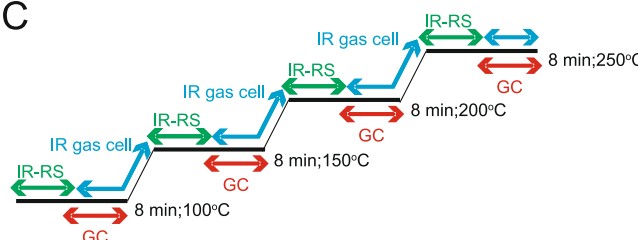

**Fig. 8 UV–Vis results and scheme of FT-IR measurements.** Diffuse-reflectance UV–Vis spectrum of dehydrated ( 480 °C) Co(II)-ZSM-5 with a high portion of Al pairs (**a**), and spectrum deconvolution to components describing Co(II) in α-, β- and γ-rings (**b**). Measurement sequence: IR-RS, rapid scan IR measurement (one spectrum collected within 1 s); GC, gas chromatography measurement; IR gas cell, one RS-IR spectrum followed by one spectrum of gaseous products collected in the PIKE IR gas cell (**c**).

**Operando spectroscopic studies**. The analysis of the mechanism of ethanol adsorption on zeolite's surface was determined by time-resolved rapid scan IR measurements in the operando rig embedded in a flow setup (RS-IR). Prior to the catalytic reaction, the catalyst wafers (35 mg) were pretreated under continuous nitrogen flow for 60 min at 550 °C. Then, nitrogen as a carrier gas was passed over the liquid ethanol (analytical grade, 99% purity), by obtaining the concentration of ethanol in helium flow at 0.56 mmol dm$^{-3}$. The total flow rate of the feed stream was kept at 30 cm$^3$ min$^{-1}$. Analysis of the ethanol conversion was initiated at RT, and followed by 100, 150, 200 and 250 °C. IR spectra were collected with a Vertex 70 Bruker spectrometer equipped with MCT detector. The spectral resolution was of 2 cm$^{-1}$. In RS-IR experiments one spectrum (30 scans) was accumulated in 1 s. The two-dimensional correlation spectroscopy was performed over recorded 1D spectra in each temperature studied. The 2D COS graphs were prepared with OPUS 3D software from Bruker Optics. Both reactants and/or the reaction products were analyzed by the infrared gas cell with MCT detector and gas chromatography. The schematic representation of the measurement step order is presented in Fig. 8c.

## Data availability

The authors declare that the data supporting the findings of this study are available within the paper.

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

## Acknowledgements

This work was financed by the National Science Centre of Poland (Grant No. 2015/18/E/ST4/00191) and Academy of Sciences of the Czech Republic (project RVO: 61388955).

## Author contributions

K.G.-M., E.T. and J.D. supervised the progress of the entire project. K.G.-M. designed all the experiments. V.P. and E.T. prepared the zeolite samples. K.G. and K.T. carried out the rapid scan IR spectrometry and operando experiments. All the authors contributed to the discussion regarding various aspects of the project. The paper was written through collective contributions from all authors. All authors approved the final version of the paper.

## Competing interests

The authors declare no competing interests.
