## [Peer review file · Communications Chemistry]

Reviewers' comments:

Reviewer #1 (Remarks to the Author):

The revised manuscript is an important added value to the knowledge in the field of the mechanism in ethanol dehydration on zeolites depending on the location of Lewis and Bronsted acid sites estimated by the location of Al atoms. Therefore it is worth of publishing, but before it the manuscript requires minor revision. Below please find my remarks.

1. The authors based on the earlier study of some of the them, on the location of Al atoms in ZSM-5 zeolite depending on the preparation procedure. In this contribution the synthesis and characterization of the prepared zeolites showing the location of Al atoms was performed according to the earlier study. It should be clearly indicated in the manuscript to show readers clearly that novelty of this paper is the use of FTIR time-resolved operando study for evaluation of the ethanol dehydration depending on the presence of isolated Al atoms vs pairs of Al atoms in the zeolite structure.
2. The description of two different reaction pathways at the beginning of the Results section is not consistent with the scheme 1. In my opinion the description is correct but the scheme contains mistakes. In the first step of the route via dissociative ads. of ethanol – ethoxy species are formed and water. Therefore, in the scheme OH bridged in the zeolite cannot still exist. This bridged OH is reconstructed in step G. Protons comes from alcohol OH group. Moreover “C” is missing in associative pathway.
3. Table 1 - Why the amount of Al is the same for both zeolites but Si/Al is different? Acidic strength needs units (not only values).
4. Line 128 - What is an experimental proof for enhanced basicity of the oxygen mentioned?
5. Line 2014 - Why here is reference to methanol dehydration?

Reviewer #2 (Remarks to the Author):

The COMMSCHEM-19-0251 manuscript is a very good paper devoted to the investigation of catalytic conversion of ethanol on two ZSM-5 zeolite materials with monomodal but different distributions of Al atoms in the zeolite structure, strictly isolated Al atoms and two Al atoms in one ring. The authors of this paper by the FT-IR time-resolved operando study supported with catalytic results have demonstrated that the reaction pathway in ethanol transformation over ZSM-5 has been controlled by the proximity of Al atoms in the framework. They have shown that ZSM-5 with the majority of isolated Al atoms transformed ethanol in the associative pathway, in contrast ZSM-5 with a dominating fraction of two Al atoms in one ring formed dimethyl ether from ethanol on the dissociative pathway. The authors revealed in the paper that the Al distribution represents a key parameter which control the reaction pathway of ethanol transformation over ZSM-5 zeolite.

This manuscript is clear and merits publication in Communications Chemistry journal in present form.

Reviewer #3 (Remarks to the Author):

The conclusions of the authors do not match with those obtained by other authors. In particular, the conclusion that "the reaction pathway in ethanol transformation over ZSM-5 is controlled by the proximity of Al atoms in the framework" is in my opinion wrong. On the other hand, the technique used for determine the distribution of aluminum ions in the zeolite framework is, in my opinion, not reliable. You have the same spectra in the case of cobalt catalysts deposited on amorphous supports. The IR work is good, but no more than hundreds of previous publications.

We thank to the Referees for their great effort to provide us with valuable recommendations to improve the manuscript. Below are given the changes done in the revised text and our comments.

Reviewers' comments:

Reviewer #1 (Remarks to the Author):

The revised manuscript is an important added value to the knowledge in the field of the mechanism in ethanol dehydration on zeolites depending on the location of Lewis and Bronsted acid sites estimated by the location of Al atoms. Therefore it is worth of publishing, but before it the manuscript requires minor revision. Below please find my remarks.

1. The authors based on the earlier study of some of the them, on the location of Al atoms in ZSM-5 zeolite depending on the preparation procedure. In this contribution the synthesis and characterization of the prepared zeolites showing the location of Al atoms was performed according to the earlier study. It should be clearly indicated in the manuscript to show readers clearly that novelty of this paper is the use of FTIR time-resolved operando study for evaluation of the ethanol dehydration depending on the presence of isolated Al atoms vs pairs of Al atoms in the zeolite structure.

Answer: We thank the Referee for this comment. The manuscript was modified to highlighted the novelty of this paper, which lies with application of FTIR time-resolved operando study to determine the influence of Al distribution on ethanol dehydration. The following text was added to manuscript:

The novelty of this work lies with the first application of FTIR time-resolved operando study for direct evidence of the effect of aluminium organization in the zeolites on the reaction pathways in acid catalysed reactions.

2. The description of two different reaction pathways at the beginning of the Results section is not consistent with the scheme 1. In my opinion the description is correct but the scheme contains mistakes. In the first step of the route via dissociative ads. of ethanol – ethoxy species are formed and water. Therefore, in the scheme OH bridged in the zeolite cannot still exist. This bridged OH is reconstructed in step G. Protons comes from alcohol OH group. Moreover "C" is missing in associative pathway.

Answer: We apologise for discrepancy between the description of mechanism in the text and Scheme 1. The mistakes in the reaction scheme were corrected and missing C part was added. The modified Scheme 1 is presented below:

Scheme 1. Dissociative and associative reaction pathways for ethanol dehydration over zeolites.

3. Table 1 - Why the amount of Al is the same for both zeolites but Si/Al is different?

Answer: We apologise for this imprecision. Previous version of Table 1 contained an average amount of Al atoms due to the close value of Si/Al in both samples was closed. We corrected the Table 1 as follows:

Table 1. Chemical composition of H-ZSM-5(s) and H-ZSM-5(p), textural properties, $[\text{Co}(\text{II})(\text{H}_2\text{O})_6]^{2+}$ ion-exchange capacity Co_{max} , percentage of Al in Al pairs or single Al sequence, and fraction of Al (H^+) located in channel intersections.

Material	Si/Al ^a	Al _{ICP} ^a	S _{micro} ^b	S _{BET} ^b	V _{micro} ^b	V _{total} ^b	Co _{max} /Al molar	Single Al	Al pairs	α^c	β^c	Al intersections (H^+) at
		$\mu\text{mol}\cdot\text{g}^{-1}$	$\text{m}^2\cdot\text{g}^{-1}$	$\text{cm}^3\cdot\text{g}^{-1}$	$\text{cm}^3\cdot\text{g}^{-1}$	%						
ZSM-5(s)	15.0	900	425	440	0.18	0.21	0.09	83	17	n.d.	n.d.	>95
ZSM-5(p)	14.5	995	404	419	0.17	0.20	0.35	30	70	37	54	~ 65

Acidic strength needs units (not only values)

Answer: The acid strength of protonic sites was determined using quantitative Py-sorption on dehydrated zeolites monitored by FTIR. The Py330 /Py170 ratio (where Py330 and Py170 represent the value of pyridine (Py) band intensity after adsorption at 330°C and 170°C, respectively) expresses the amount of pyridine molecules neutralizing protonic sites that survived desorption at 330 °C (Table S2). Thus, the acid strength is presented as a

dimensionless value. However, according to the Referee suggestion the comment describing the pyridine sorption was added to the manuscript.

4. Line 128 - What is an experimental proof for enhanced basicity of the oxygen mentioned?

Answer: In zeolites materials, the negative charge induced by framework Al^{3+} cations, is compensated by extra-framework mono- or multivalent cations. The negative charge transfer by oxygen framework atoms primarily depends on the structure and chemical composition of the zeolite. A simple but powerful concept allowing assessing the average negative charge of framework oxygen atoms vs. composition is the electronegativity equalization principle of Sanderson [R. T. Sanderson, *Science*, 1951, 114, 670., R. T. Sanderson, *Chemical Bonds and Bond Energy*, Academic Press, New York, 1976] which was applied to zeolites by Mortier [W. J. Mortier, *J. Catal.*, 1978, 55, 138.] Within this paradigm [S. Bordiga, C. Lamberti, F. Bonino, A. Travert, F. Thibault-Starzyk “Probing zeolites by vibrational spectroscopies” *Chem. Soc. Rev.*, 44 (2015) 7262-7341], the average partial charge of the framework oxygen atoms δ_{O} is proportional to the difference between the intermediate electronegativity of the zeolite atoms, S_{int} with the oxygen electronegativity ($S_{\text{O}} = 5.21$):

$$\delta_{\text{O}} \propto S_{\text{int}} - S_{\text{O}}$$

where S_{int} is the geometric mean of the electronegativities of the component atoms of the zeolite. In other words, the lower the electronegativities of the cations, the lower the intermediate electronegativity, the more negative the oxygen framework atoms.

Hence, the oxygen charge increases with the amount of Al in the framework ($S_{\text{Al}} = 2.22$ vs. $S_{\text{Si}} = 2.84$), or when the electronegativity of the extra-framework counter-cations is low (for instance for alkaline-exchanged zeolites, the oxygen charge increases from Li to Cs-exchanged zeolites: $S_{\text{Li}} = 0.74$, $S_{\text{Cs}} = 0.28$).

5. Line 2014 - Why here is reference to methanol dehydration?

Answer: We thank the Referee for the comment. The reference was added to support the role of confinement effect on 10-rings zeolites as model molecule methanol was used.

Reviewer #2 (Remarks to the Author):

The COMMSCHEM-19-0251 manuscript is a very good paper devoted to the investigation of catalytic conversion of ethanol on two ZSM-5 zeolite materials with monomodal but different distributions of Al atoms in the zeolite structure, strictly isolated Al atoms and two Al atoms in one ring. The authors of this paper by the FT-IR time-resolved operando study supported with catalytic results have demonstrated that the reaction pathway in ethanol transformation over ZSM-5 has been controlled by the proximity of Al atoms in the framework. They have shown that ZSM-5 with the majority of isolated Al atoms transformed ethanol in the associative pathway, in contrast ZSM-5 with a dominating fraction of two Al atoms in one ring formed dimethyl ether from ethanol on the dissociative pathway. The authors revealed in the paper that the Al distribution represents a key parameter which control the reaction pathway of ethanol transformation over ZSM-5 zeolite.

This manuscript is clear and merits publication in Communications Chemistry journal in present form.

Reviewer #3 (Remarks to the Author):

The conclusions of the authors do not match with those obtained by other authors.

Answer: We strongly disagree. Although there is only limited number of papers on the effect of Al distribution on acid catalysed reactions, its significant effect on the catalytic performance was observed. Davis (ACS. Catal., 6, 2016, 542-550), Gounder (ACS. Catal., 7, 2017, 6663-6674) and Corma (Gallego et al., Chem.-Eur. Journal, 24, 2018, 14631-14635) clearly showed that selectivity and activity of methanol to olefins transformation over SSZ-13 zeolite is dramatically affected by Al distances in this matrix. Note that SSZ-13 zeolite is of CHA topology and its framework is build using only one framework T atom. Thus, other parameters of Al organization cannot play a role.

In particular, the conclusion that "the reaction pathway in ethanol transformation over ZSM-5 is controlled by the proximity of Al atoms in the framework" is in my opinion wrong.

Answer: The opinion of Reviewer #3 is strictly contradicted with the experimental data reported in the paper. Even not taking in account spectroscopic characterization of samples, both samples dramatically differ in the maximum ion exchange capacity of the zeolites for Co(II) ions. All acid sites of protonic forms of catalysts are accessible for probe molecules and samples do not exhibit the presence of extra-framework Al species. Thus, channel system is not

blocked and differences in the ion exchange capacity reflect differences in Al distribution (single Al atoms do not accommodate divalent Co(II) species). The results of FTIR experiments and catalytic tests clearly evidence different catalytic performance of investigated samples, which evidenced a significant difference in Al distribution, while other parameters are similar. Thus, it is clearly indicating that differences in catalytic activity is related with Al distribution.

On the other hand, the technique used for determine the distribution of aluminum ions in the zeolite framework is, in my opinion, not reliable.

Answer: We completely disagree with the Reviewer #3. Analysis of Al distribution is discussed in several reviews on this subject (Dedecek at al., *Catal. Rev.-Sci.Eng.*, 54, 2012, 135-223; Gounder et al., *ACS, Catal.*, 8, 2018, 770-784, and and Dedecek at al., *Chem. Sus. Chem.*, 12, 2019, 556 – 576) and is employed by several laboratories worldwide:

1) Presence of single Al atoms and Al pairs was recently confirmed by their direct evidenced for ZSM-5 using ^{27}Al - ^{27}Al double-quantum single-quantum MAS NMR (Taulelle et al., *Solid State NMR*, 84, 2017, 65-72). Note that samples used in this NMR study are analogues of those used in the manuscript.

2) Various indirect methods for the analysis of Al distribution in zeolites were developed since this parameter was introduced to the authors to the zeolite science.

2a) Only maximum ion exchange capacity of the zeolite for divalent cations (titration of Al pairs) can serve as a measure of Al distribution. Limits of this approach are discussed in the detail in (Dedecek at al., *Chem. Sus. Chem.*, 12, 2019, 556 – 576).

2b) Bare Co(II) ions in dehydrated zeolites can be monitored not only using UV-Vis spectroscopy, but also using FTIR spectroscopy of the perturbation of anti-symmetric T-O-T vibrations of the zeolite ring due to the ligation of bare Co(II) or other M(II) ions. It has been pointed out, that FTIR spectroscopy of T-O-T vibrations is based on different physical principle than UV-Vis spectroscopy. However, a sound agreement between both methods was reported in the case of ZSM-5, ferrierite and mordenite zeolites (Prins et al., *J. Catal.*, 194, 2000, 330-342 and *J. Phys. Chem. B*, 106, 2002, 2240-2248).

2c) Gounder and his group developed approach based on the titration of Al pairs by Co(II) in the combination with the study of residual not ion exchanged protons using NH_4 TPD (Gounder et al., *Chem. Mater.*, 28, 2016, 2236-2247; *ACS. Catal.* 7, 2017, 6663-6674, and *J. Phys. Chem. C*, 123, 2019, 17454-17458) and his results obtained for SSZ-13 wits well with our results

obtained by the combination of Co(II) UV-vis and FTIR spectroscopies (Dedecek et al., J. Phys. Chem. C, 123 (2019) 7968–7987).

2d) Davis developed own method for the characterization of Al distribution based instead of Co(II) ions on Cu(II) ions (Davis et al., ACS Catal. 2016, 6, 542-550).

You have the same spectra in the case of cobalt catalysts deposited on amorphous supports.

Answer: Reviewer #3 has been wrong. Cobalt (II) ions monitored by Vis spectroscopy were successfully employed for the analysis of Al distribution and siting of Co(II) ions not on our by also by other laboratories as e.g. Berkeley (A. Janda and A. T. Bell , J. Am. Chem. Soc., 2013, **135** , 19193 -19207), ETH Zurich (van Bokhoven et al. J. Catal., 2001, **202** , 129 -140,), KU Leuven (Schoonheydt et al., J. Phys. Chem., 99, 1995, 15222-15228). In his statement, Reviewer #3 ignores the fact, that in the crystalline matrix of the zeolite, the number of cationic sites (rings) of Co(II) ions is strictly limited, as only 6-member rings or in some exceptions 8-member (mordenite, SSZ-13) rings can accommodate bare divalent cations. For many zeolites (ferrierite, mordenite, SSZ-13 of CHA topology), these cationic sites are well known from X-ray diffraction and fits well to UV-Vis results. It should be pointed out, that in the case of cation siting in the ferrierite zeolite, cationic sites for divalent cations were suggested based on our UV-Vis results and only confirmed two years later by X-ray diffraction results. Moreover, the interpretation of Co(II) spectra presented by Reviewer #3 is not correct.

1) In the case of Co(II) on amorphous surfaces, perturbed tetrahedrally coordinated Co(II) ions are typically present on amorphous matrices and exhibit asymmetric triple centered around $17\ 000\ \text{cm}^{-1}$ which is very closed to the complex spectrum of Co(II)-zeolites. However, in the case of tetrahedral Co(II) coordination, analogous triplet with identical splitting has to be (and is) observed in the near IR region. On contrary, in the case of bare Co(II) ions in cationic positions in zeolites, their coordination is far from the tetrahedral one. Planar or close to planar coordination of Co(II) ion and low concentration of pseudo octahedral Co(II) ions is observed. In this case, the Vis spectrum is not followed by its analogue with identical splitting in NIR region due to quite different structure of the d-d transitions.

2) Under specific conditions, Co(II) ions with the co-ordinations (approximately planar coordinated to three or four oxygens) similar to those reported for zeolites can be maybe prepared (there is not many possibilities for Co(II) coordination on the surface of amorphous

silica or alumina). In this case, spectra should be similar to those of zeolites and probably even employed for the characterization of local coordination of Co(II) on amorphous surfaces. Spectra of Co(II) in zeolites can in this case serve as finger prints of individual co-ordinations of Co(II) ions.

The IR work is good, but no more than hundreds of previous publications

Answer: We strongly disagree with the statement that “*The IR work is good, but no more than hundreds of previous publications*”. According to my best knowledge supported by many years of my and co-authors experience in the field of application of an IR spectroscopy to solid state, only a few papers showed the application of two-dimensional correlation rapid scan IR spectroscopy (2D COS RS IR spectroscopy), for characterization of the working heterogeneous catalyst. Some works comes from Laboratoire Catalyse et Spectrochimie in Caen (France) (F. Thibault-Starzyk et al. 2D-COS IR study of coking in xylene isomerization on H-MFI zeolite, *Catalysis Today* 70 (2001) 227–241; Ch. Fernandez, et al., Hierarchical ZSM-5 Zeolites in Shape-Selective Xylene Isomerization: Role of Mesoporosity and Acid Site Speciation; *Chem. Eur. J.* 2010, 16, 6224 – 6233;) other more recent papers originates from my group (K. Gołabek, *Dalton Transactions* 46 (2017) 9934–9950, K.A.Tarach et al. *Catal. Today*, 283(2017)158, K. Gołabek et al. *Microporous Mesoporous Mater.* 266 (2018) 90, K. A. Tarach et al., *Applied Catalysis A, General* 568 (2018) 64–75; K. Gołabek et al. *Spectrochimica Acta Part A: Molecular and Biomolecular Spectroscopy* 192 (2018) 464–472). Those papers are focused on xylenes transformations over zeolitic catalysts but consider also the ethanol conversion on zeolites of various 10-ring topology. To our best knowledge, in the reviewed paper, for the first time we discussed the results received by combination of the possibilities served by rapid scanning and 2D COS IR spectroscopy. Moreover, the implementation of such methodology for investigation of subtle effects occurring during ethanol transformation over differentiated but monomodal Al distribution in the framework have never been studied before. *The novelty of this study lies with application of FTIR time-resolved operando study for evaluation of the reaction pathways of ethanol dehydration over isolated Al atoms or Al atoms pairs in MFI structure.*

In addition, there are no reports evidencing that the reaction pathway in ethanol transformation over ZSM-5 is controlled by the proximity of Al atoms in the framework. Till now, there was no direct spectroscopic proof for formation of ethoxy or ethanol dimeric species, their presence

in the reaction pathways were only postulated. Our breakthrough results are easily applicable for other adsorbate-zeolitic support systems as the confine sorption, thus reaction pathway is directed by the Al atoms distribution. We evidenced that on-purpose tuned distribution of Al atoms opens the door for the formation of highly selective tailor-made zeolitic catalysts by the tuning of the type of the surface intermediates formed.

Reviewers' comments:

Reviewer #1 (Remarks to the Author):

The authors revised the manuscript according to my remarks. Thus, I accept the revised manuscript.

Reviewer #3 (Remarks to the Author):

I confirm my opinion. The experimental work is good but not better than most already published work. UV-vis spectroscopy does not give any precise information on the position of cobalt II in zeolites. The conclusions about ethanol conversion mechanism are wrong. The paper, written in this way, must be rejected.

Reviewer #4 (Remarks to the Author):

This paper describes an attempt to demonstrate the effect of different Al locations in similar ZSM-5 zeolites. The work is based on the use of NMR and Co(II) ion exchange methods to estimate the location and number of Al pairs and then extrapolate this information to explain the catalytic mechanism of ethanol transformations.

There is clearly a disagreement between the referees as to the importance of this work. I looked at the referee comments and then explored three main areas of the published paper:

1. Is the basic characterisation of the materials sufficient to reach the conclusions?
2. Is the catalytic work enough to support the conclusions?
3. Is the clarity of the manuscript sufficient to explain the conclusions to the reader?

Referee 3 has some reservations about the basic technique to determine the number of Al pairs. The basic characterisation of the materials, in particular the calculation of the number of Al pairs, is done pretty much according to the recent literature and this seems to have become relatively well accepted, and so in that sense the author are correct. I do have some sympathy with referee 3 in his comment about the unreliability of the method in the sense that I don't believe that the repeatability has been proved beyond doubt. I would like to see the authors put some sensible error estimates on the numbers based on their experiments. However, apart from that there is enough in the literature to suggest that the method is at least broadly appropriate. What is clear is that there is an effect on the catalysis and so I will look at that next.

2. The main criticism is that the work disagrees with other results from different groups and that the mechanism is simply wrong. In reviewing this paper I have concentrated on looking at the previous work by Gounder as this seems the most comprehensive series of papers especially on CHA type zeolites. There are certainly similarities in the results he presents in certain areas (e.g. the dominant dissociative pathway for methanol transformations). Therefore I don't really agree with the referee about the lack of precedent from other groups. The basic logic seems similar and so I am broadly happy with the conclusions presented.

3. The clarity and precision of the paper itself is, however, not very good. It took me a long time to decipher what the authors were trying to say in quite a lot of the paper. As examples one only has to look at the Figures.

I paste the caption to Figure 3

"Fig. 3. Schematic depictions of single Al (A) and Al pairs (B) in a zeolite framework; Al atoms in red. IR spectra of ZSM-5 studied after 60 min at 550 oC followed by 10 min at 100 oC in N₂ (red line), 20 s interaction with ethanol at 100 oC (blue line) and upon the contact with ethanol on increasing temperature from 100 oC to 250 oC within 4 min (black lines)."

first, the figure has no A and B marked on it. Instead it has A and C (but there is no explanation of what these are - except considerably later in the text). There is a red line on only half the spectra! Are they hiding something?

This is symptomatic of most of the figures - in Figure 4 on one side of the lower graph it talks about component C, but on the other side it does not mention if it is supposed to be component -

Figure 5 - the text talks of negative correlations but there is no scale - which are negative? In the text an important peak is the 3530, but only 3550 is marked on the figures

Figure 6 - the legend shows DEE but this is not explained - in the text it talks about production of dimethyl ether - something isn't right.

The text could also do with a careful re-reading. There are some sentences which don't actually say what I think the authors are trying to say!

For the reader though, perhaps the most difficult thing is the relationship between Figure 2 and the catalytic results in Figure 6. Clearly, this is not straightforward but one of the possible pathways does not show how the dominant product can be formed.

All this leads to difficulty in determining whether the research has been reliably completed, and so I have much sympathy with how referee 3 came to their opinion.

I think the IR does show what the authors are claiming and so I am broadly positive about the work in general, but the paper itself needs to be tightened up before it can be accepted, and there is a particular need to focus on the detail of the presentation.

We thank to the Reviewer #4 for time spending for reading the manuscript and for his/her great effort to provide us with valuable recommendations to improve the manuscript. Below are given the changes done in the revised text and our comments.

Reviewer #4 (Remarks to the Author):

This paper describes an attempt to demonstrate the effect of different Al locations in similar ZSM-5 zeolites. The work is based on the use of NMR and Co(II) ion exchange methods to estimate the location and number of Al pairs and then extrapolate this information to explain the catalytic mechanism of ethanol transformations.

There is clearly a disagreement between the referees as to the importance of this work. I looked at the referee comments and then explored three main areas of the published paper:

1. Is the basic characterisation of the materials sufficient to reach the conclusions?
2. Is the catalytic work enough to support the conclusions?
3. Is the clarity of the manuscript sufficient to explain the conclusions to the reader?

Referee 3 has some reservations about the basic technique to determine the number of Al pairs. The basic characterisation of the materials, in particular the calculation of the number of Al pairs, is done pretty much according to the recent literature and this seems to have become relatively well accepted, and so in that sense the author are correct. I do have some sympathy with referee 3 in his comment about the unreliability of the method in the sense that I don't believe that the repeatability has been proved beyond doubt. I would like to see the authors put some sensible error estimates on the numbers based on their experiments. However, apart from that there is enough in the literature to suggest that the method is at least broadly appropriate. What is clear is that there is an effect on the catalysis and so I will look at that next.

Answer: The inaccuracy of the analysis of the concentration of Al in pairs is below ca. 10 rel. %. There are in general four methods for the estimation of the concentration of Al pairs in the zeolite:

1. Vis spectroscopy of the d-d transitions of the Co(II) ions in the dehydrated zeolite, which was employed in the paper. One bare Co(II) balances one pair of Al atoms in individual cationic site. Vis spectra are simulated using Gaussian band and concentration of Co(II) ions in individual sites is calculated using already published extinction coefficient (Dedecek et al., Micropor. Mesopor. Mater. 35-36 (2000) 483-494).
2. FTIR spectroscopy of antisymmetric T-O-T skeletal vibrations. In this case, ligation of the bare Co(II) ion to the ring with Al pair results in a new T-O-T band. Simulation of the spectrum (significantly more simple compared to the simulation of the Vis spectra due to the fact that only one band corresponds to one cationic site) and application of extinction

coefficient (the same for all sites –Al pairs) gives concentration of Al pairs (most recently in Lemishka et al., J. Pure Appl. Chem., <https://doi.org/10.1515/pac-2018-1228>).

3. FTIR spectroscopy of d3-acetonitrile adsorbed on bare Co(II) ions. In this case, bare Co(II) ions play a role of a Lewis acid site and there is not possible to distinguish siting in different sites (different Al pairs). Extinction coefficient for ZSM-5 was not published yet.

4. Indirect estimation from Co(II) exchange capacity. Co(II) exchange capacity is a direct proof of the concentration of single Al atoms as also close unpaired Al atoms can be present in zeolites (Beta, SSZ-13). Nevertheless, in ZSM-5, the close unpaired Al atoms (reflected in the formation of Co-oxo species in dehydrated zeolite) were never observed. Thus, Co(II) maximum loading can be also taken as a measure of Al pairs.

Although it was not published yet, for a number of samples of several structural types of zeolites (for our knowledge and for commercial partners) during ca. 15 years of the research we employed combination of all above methods for the analysis of Al distribution. As follows from the comparison of all methods, in the case of samples with lower amount of Al pairs (high fraction of pairs in Si rich samples ($\text{Si}/\text{Al} > 20$) or low fraction of Al pairs and $\text{Si}/\text{Al} < 20$), the inaccuracy of the analysis was below 10 rel. %. In the case of samples rich with Al pairs and $\text{Si}/\text{Al} < 17$, the accuracy of the analysis was about 5 rel. %.

It has been pointed out that we are able to reach accuracy below 15 rel. % even in the reproducibility of the ZSM-5 samples with controlled Al distribution, i.e. of the synthesis of the zeolite and subsequent analysis of Al distribution.

2. The main criticism is that the work disagrees with other results from different groups and that the mechanism is simply wrong. In reviewing this paper I have concentrated on looking at the previous work by Gounder as this seems the most comprehensive series of papers especially on CHA type zeolites. There are certainly similarities in the results he presents in certain areas (e.g. the dominant dissociative pathway for methanol transformations). Therefore I don't really agree with the referee about the lack of precedent from other groups. The basic logic seems similar and so I am broadly happy with the conclusions presented.

3. The clarity and precision of the paper itself is, however, not very good. It took me a long time to decipher what the authors were trying to say in quite a lot of the paper. As examples one only has to look at the Figures.

First, the figure has no A and B marked on it. Instead it has A and C (but there is no explanation of what these are - except considerably later in the text). There is a red line on only half the spectra! Are they hiding something?

Answer: We are sorry for this oversight. The line "Schematic depictions of single Al (A) and Al pairs (B) in a zeolite framework; Al atoms in red." Referring to Fig. 1 was removed.

The A and C notation is now given in caption to Fig. 3.

Indeed, in the 1800 – 1380 cm^{-1} region the difference spectrum of activated zeolite was not presented for clarity. It is a flat, straight line (as the spectrum of activated was subtracted from itself). However, according to Reviewer suggestion it was implemented in Fig. 3. The caption of Fig. 3 is now as below:

"Fig. 3. Schematic depictions of single Al (A) and Al pairs (B) in a zeolite framework; Al atoms in red. IR spectra of ZSM-5 studied after 60 min at 550 oC followed by 10 min at 100 oC in N₂ (red line), 20 s interaction with ethanol at 100 oC (blue line) and upon the contact with ethanol on increasing temperature from 100 oC to 250 oC within 4 min (black lines). The component C is presented as the difference between the spectrum collected in the sequence described above and the spectrum of activated zeolite "

This is symptomatic of most of the figures - in Figure 4 on one side of the lower graph it talks about component C, but on the other side it does not mention if it is supposed to be component.

Answer: We improved the manuscript in line with the Reviewer suggestions. To make the results discussion more clear description of the data analyzed in the pictures was explained in the manuscript.

Figure 5 - the text talks of negative correlations but there is no scale - which are negative? In the text an important peak is the 3530, but only 3550 is marked on the figures.

Answer: The scale presenting the correlation levels were added to the Fig. 5. The proper explanation was also implemented to the revised manuscript (page):

On 2D maps the overlapping peaks are well-resolved what allow their detecting and characterising the specific sequential order of spectral intensity changes. The 2D synchronous map presenting correlations between ν_1 and ν_2 wavenumbers consist of the positive autopeaks and the positive or negative crosspeaks. The positive crosspeaks represent the simultaneous changes in the intensities of the ν_1 and ν_2 bands, i.e. their simultaneous increase or decrease. The negative crosspeaks represent the band intensity changes at the expense of one to the other.

Figure 6 - the legend shows DEE but this is not explained - in the text it talks about production of dimethyl ether - something isn't right.

Answer: We thank the Reviewer for his/her comment. The term "dimethyl" was substituted for the correct one: "diethyl".

For the reader though, perhaps the most difficult thing is the relationship between Figure 2 and the catalytic results in Figure 6. Clearly, this is not straightforward but one of the possible pathways does not show how the dominant product can be formed.

Answer: Fig. 2 presenting the characterization of the studied samples was added to the manuscript to confirm the structural properties of synthesized ZSM-5(p) and ZSM-5(s). FTIR study of adsorbed probe molecule were performed to check whether the presence

of close protons in ZSM-5(p) did not influence the strength acidity. Presented results showed that both samples exhibited the same strength acidity and differing only in Al organization in the zeolite framework, thus they can be compared in the study of ethanol dehydration. Fig. 6 depicted the catalytic results of transformation of ethanol over ZSM-5(p) and ZSM-5(s), which confirmed that depending of the Al distribution in ZSM-5 matrix different products are formed.

The text could also do with a careful re-reading. There are some sentences which don't actually say what I think the authors are trying to say!

All this leads to difficulty in determining whether the research has been reliably completed, and so I have much sympathy with how referee 3 came to their opinion. I think the IR does show what the authors are claiming and so I am broadly positive about the work in general, but the paper itself needs to be tightened up before it can be accepted, and there is a particular need to focus on the detail of the presentation.

Answer: According to Reviewer suggestion manuscript was carefully re-reading and improving. We really tried to make the manuscript more reader friendly.

REVIEWERS' COMMENTS:

Reviewer #4 (Remarks to the Author):

The authors have made a good effort to address the points I raised. I am now happy with the manuscript